# Does Nut Consumption Reduce Mortality and/or Risk of Cardiometabolic Disease? An Updated Review Based on Meta-Analyses

**DOI:** 10.3390/ijerph16244957

**Published:** 2019-12-06

**Authors:** Yoona Kim, Jennifer B Keogh, Peter M Clifton

**Affiliations:** 1Department of Food and Nutrition/Institute of Agriculture and Life Science, Gyeongsang National University, Jinju 52828, Korea; yoona.kim@gnu.ac.kr; 2School of Pharmacy and Medical Sciences, University of South Australia, General Post Office Box 2471, Adelaide, SA 5001, Australia; jennifer.keogh@unisa.edu.au

**Keywords:** meta-analyses, nuts, cardiometabolic disease, low-density lipoprotein cholesterol, fasting blood glucose

## Abstract

*Aim* We aimed to determine if nut consumption decreases mortality and/or the risk of cardiometabolic diseases based on updated meta-analyses of epidemiological and intervention studies. Methods. An updated electronic search was conducted in PubMed/MEDLINE, Cumulative Index to Nursing and Allied Health Literature (CINAHL), and the Cochrane Library databases for original meta-analyses to investigate the effects of nut consumption on cardiometabolic disease in humans. Results. Seven new meta-analyses were included in this updated review. Findings similar to our previous review were observed, showing that nut consumption significantly decreased cardiovascular disease (CVD) mortality (−19% to −25%; *n* = 4), coronary heart disease (CHD) mortality (−24% to −30%; *n* = 3), stroke mortality (−17% to −18%; *n* = 3), CVD incidence (−15% to −19 %; *n* = 4), CHD [or coronary artery disease (CAD)] incidence (−17% to −34%; *n* = 8), and stroke incidence (−10% to −11%; *n* = 6) comparing high with low categories of nut consumption. Fasting glucose levels (0.08 to 0.15 mmol/L; *n* = 6), total cholesterol (TC; 0.021 to 0.30 mmol/L; *n* = 10), and low-density lipoprotein cholesterol (LDL-C; 0.017 to 0.26 mmol/L; *n* = 10) were significantly decreased with nut consumption compared with control diets. Body weight and blood pressure were not significantly affected by nut consumption. Conclusion. Nut consumption appears to exert a protective effect on cardiometabolic disease, possibly through improved concentrations of fasting glucose, total cholesterol, and LDL-C.

## 1. Introduction

Nuts comprise 43–67% fat and 8–22% protein by weight. Nuts have unique nutritional profiles. Nuts are abundant in unsaturated fatty acids (UFAs) (containing both monounsaturated fatty acids (MUFAs) and polyunsaturated fatty acids (PUFAs)) and only 4–5% of saturated fatty acids (SFAs). Nuts are high in vitamins, minerals, fiber, and bioactive compounds, including carotenoids, phytosterols, and polyphenols [1,2].

As we suggested, for a potential mechanism of action of nuts [3], MUFAs and PUFAs are candidates for favorable glucose control and reduction of appetite. Arginine and magnesium can contribute to improved inflammation, oxidative stress, endothelial function, and blood pressure. Polyphenols can lower the risk of type 2 diabetes mellitus (T2DM). The suggested schematic figure showing metabolic effects and effects on clinical endpoints based on our previous publication [3] is briefly described in Figure 1. 

As previously reported [4], nut consumption was inversely associated with all-cause mortality (−19% to −20%; *n* = 6), cardiovascular disease (CVD) mortality (−25%; *n* = 3), coronary heart disease (CHD) mortality (−27% to −30%; *n* = 2), and stroke mortality (−18%; *n* = 2). Moreover, nut consumption was associated with lowered risks of CVD (−19%; *n* = 3), CHD (−20% to −34%; *n* = 2), stroke (−10% to −11%; *n* = 7), and hypertension (−15%; *n* = 3) from meta-analyses of prospective studies. No effect of nut consumption on T2DM risk was observed in prospective studies, while significantly lowered fasting glucose levels (−0.08 to −0.15 mmol/L) were observed in meta-analyses of randomized control trials (RCTs). In meta-analyses of RCTs, decreases in total cholesterol (TC; (−0.021 to −0.28 mmol/L) and low-density lipoprotein cholesterol (LDL–C; −0.017 to −0.26 mmol/L) and improvements in endothelial function (0.79% to 1.03% increase in flow-mediated dilation) were observed. No effects on body weight, inflammatory markers, and blood pressure were seen with nut consumption.

This review aims to update our previous review of meta-analyses in order to determine the effect of nut consumption on cardiometabolic disease.

## 2. Materials and Methods

The literature search was conducted in PubMed/MEDLINE, Cumulative Index to Nursing and Allied Health Literature (CINAHL), and the Cochrane Library databases, restricted to full articles investigating meta-analyses of the effects of nut consumption on cardiometabolic diseases in humans. Only research articles written in English up to 11 November 2019 were included. The search terms included meta-analysis combined with nut(s) or tree nut(s) or almond(s) or Brazil nut(s) or cashew nut(s) or hazelnut(s) or macadamia(s) or peanut(s) or pistachio(s) or walnut(s) or mortality or incidence or CVD or coronary heart disease (CHD) or stroke or T2DM or hypertension or metabolic syndrome or obesity or blood pressure or glycemic control or glucose or lipids or inflammatory markers or endothelial function and flow-mediated dilation. Reference lists of chosen articles were also screened for related publications. A previous review [4] examined 34 meta-analyses, and this review added 7 new meta-analyses [5,6,7,8,9,10,11]. A flow chart for the identified studies is included in this review in Figure 2.

## 3. Results

### 3.1. CVD Mortality

In the present review, one meta-analysis of prospective studies conducted by Becerra-Tomas et al., 2019 [8] was included. They [8] showed nut consumption decreased CVD mortality (413,727 subjects and 14,475 cases) by 23% (relative risk (RR) = 0.77; 95% confidence interval (CI) 0.72, 0.82; I^2^ = 3%; *p*_heterogeneity_ = 0.42) in a meta-analysis of 14 prospective studies (9 publications [12,13,14,15,16,17,18,19,20] comparing high vs low nut consumption categories). This reduction was similar to that shown in other meta-analyses. In the previous review [4], three meta-analyses [21,22,23] of prospective studies reported a 19–25% lower rate of CVD mortality comparing the highest and lowest consumptions. A 19–25% reduction in CVD mortality was seen in these 4 meta-analyses [8,21,22,23].

### 3.2. Coronary Heart Disease Mortality

A previous review [4] reported a 27–30% reduction in coronary heart disease (CHD) mortality from 2 meta-analyses of prospective studies [21,23]. When combined with the outcomes from a meta-analysis by Becerra-Tomas et al., 2019 [4], a reduction in CHD mortality ranged from 24% to 30%. 

Becerra-Tomas et al., 2019 [8] recently showed that nut consumption lowered CHD mortality (396,041 subjects and 7877 CHD deaths) by 24% (RR = 0.76; 95% CI 0.67, 0.86; I^2^ = 46%; *p*_heterogeneity_ = 0.04) in a meta-analysis of 12 prospective studies (8 publications [12,13,14,15,19,24,25,26]) comparing high with low nut consumption categories. The meta-analysis of Chen et al., 2017 [23] included 13 studies (10 publications) [13,14,15,16,17,18,19,26,27,28].

### 3.3. Stroke Mortality

Becerra-Tomas et al., 2019 [8] showed a lower risk of stroke mortality (351,618 subjects and 2332 cases) following nut consumption with a RR of 0.83 (95% CI 0.75, 0.93; I^2^ = 0%; *p*_heterogeneity_ = 0.45) from a meta-analysis of 11 prospective studies (7 publications [12,13,14,18,19,26,29]) in a comparison of highest with lowest total nut consumption. This finding is consistent with the 18% reduction from 2 previous meta-analyses [21,23].

### 3.4. Cardiovascular Disease Incidence

A previous review [5] reported nut consumption reduced CVD incidence by 19% in 3 meta-analyses [21,30,31] of prospective studies comparing high with low categories of nut consumption. When the recent meta-analysis by Becerra-Tomas et al., 2019 [8] was added [8,21,30,31], a similar reduction in incidence ranging from 15–19% was observed. Becerra-Tomas et al., 2019 [8] showed that nut consumption lowered incidence of CVD by 15% (RR = 0.85; 95% CI 0.80, 0.91; I^2^ = 0%; *p*_heterogeneity_ = 0.81) when 3 prospective studies (including the publication of Guasch-Ferre et al., 2017 [12]) from the Health Professionals Follow-up Study (HPFS), the Nurses’ Health Study I (NHSI), and Nurses’ Health Study II (NHSII) (210,836 subjects and 14,136 cases) were analyzed comparing ≥2 servings/week versus never or almost never categories of nut consumption. However, peanut butter consumption was not associated with incidence of CVD (RR = 0.98; 95% CI 0.93, 1.03; I^2^ = 89%; *p*_heterogeneity_ < 0.01). This publication [12] was only included by Becerra-Tomas et al., 2019 [8].

### 3.5. Coronary Heart Disease Incidence

A previous review [5] reported nut consumption decreased incidence of CHD (or CAD) by 17–34% in 7 meta-analyses [21,30,31,32,33,34,35] of prospective studies. When the meta-analysis by Becerra-Tomas et al., 2019 [8] was added [8,21,30,31,32,33,34,35], a 17–34% reduction in CHD (or CAD) incidence was seen. Becerra-Tomas et al., 2019 [8] showed that nut consumption lowered incidence of CHD (275,812 subjects and 12,654 cases) by 18% (RR = 0.82; CI 95% 0.69, 0.96; I^2^ = 74%; *p*_heterogeneity_ < 0.01) in a meta-analysis of 7 prospective studies (5 publications [12,24,25,36,37]) comparing high with low categories of nut consumption.

### 3.6. Stroke Incidence

A previous review [5] reported nut consumption reduced stroke incidence by 10–11% based on a meta-analysis of 10 studies (9 publications [13,14,19,26,29,38,39,40,41]) that was conducted by Aune et al., 2016 [22]. Their meta-analysis [22] had the greatest number of studies compared with other meta-analyses [21,23,25,30]. In this present review, we added a new meta-analysis conducted by Becerra-Tomas et al., 2019 [8]. They [8] showed that nut consumption was not associated with stroke incidence, but they only included 7 studies, so the findings of Aune are the most persuasive.

### 3.7. Body Weight

A previous review [5] reported no effect of nut consumption on body weight from 4 meta-analyses of prospective and intervention studies [42,43,44,45]. In this review, we included 3 more meta-analyses of observational and interventional studies [5,6,7]. As a result, nut consumption still did not significantly affect body weight based on 7 meta-analyses [5,6,7,42,43,44,45]. 

In a meta-analysis of 3 prospective studies [46,47,48] conducted by Schlesinger et al., 2019 [5], the association between nut consumption and the risk of overweight and obesity was found with an RR of 0.91 (95% CI 0.80, 1.03; I^2^ = 25%) for the highest versus lowest nut consumption. The RR per 28 g/d of nut consumption was 0.78 (95% CI 0.58, 1.06; I^2^ = 64%).

### 3.8. Randomized Controlled Trials

Very recent meta-analysis of RCTs conducted by Akhlaghi et al., 2018 [6] reported a significantly suppressed hunger (mean difference (MD) = −6.54 mm visual analogue scale (VAS); 95% CI 12.7, 0.42 mm VAS; *p* = 0.03) from a meta-analysis of 14 RCTs (12 publications [49,50,51,52,53,54,55,56,57,58,59,60]). However, no association between nut consumption and fullness (MD = 0.03 mm VAS; 95% CI 12.2, 12.3 mm VAS; *p* = 1) was observed from a meta-analysis of 10 RCTs (9 publications [49,50,51,53,54,55,57,58,61]). No association between nut consumption and weight (MD = 0.09 kg; 95% CI 0.59, 0.41 kg; *p* = 0.72) was observed from a meta-analysis of 15 RCTs (14 publications [49,53,58,59,60,62,63,64,65,66,67,68,69,70]). 

Nut consumption increased energy intake (MD = 76.3 kcal; 95% CI 22.7, 130 kcal; *p* = 0.005) from a meta-analysis of 23 RCTs (21 publications [51,53,56,58,59,60,62,63,64,65,66,67,68,69,71,72,73,74,75,76,77]). Subanalysis showed that increased energy intake following nut consumption was observed only in overweight and obese subjects, not in normal weight subjects [6]. 

Lee-Bravatti et al., 2019 [7] conducted a meta-analysis of 11 RCTs [58,60,66,78,79,80,81,82,83,84,85] (432 subjects) and showed a significant decrease in body weight after almond consumption compared with control (summary net change: −1.39 kg; 95% CI −2.49, −0.30 kg; I^2^ = 0%, *p*_heterogeneity_ = 0.87). However, almond consumption did not affect body mass index (summary net change: −0.33 kg; 95% CI −1.08, −0.43 kg; I^2^ = 21%, *p*_heterogeneity_ = 0.28), compared with controls from a meta-analysis of RCTs [58,60,66,78,79,86].

### 3.9. Glycemic Control

In a previous review [5], nut consumption was found to decrease fasting glucose levels by 0.08 to 0.15 mmol/L compared with control diets based on 3 meta-analyses [87,88,89] of RCTs. 

In this present review, we have included very recent meta-analyses conducted by Tindall et al., 2019 [9], Lee-Bravatti et al., 2019 [7], and Schwingshackl et al., 2018 [10].

Tindall et al., 2019 [9] found no association between nut consumption and fasting glucose (weighted mean difference (WMD)—0.52 mg/dL (0.028 mmol/L); 95% CI −1.43, 0.38 mg/dL; I^2^ = 53.4%) from a meta-analysis of 39 RCTs [58,61,63,66,67,68,74,78,80,83,86,90,91,92,93,94,95,96,97,98,99,100,101,102,103,104,105,106,107,108,109,110,111,112,113,114,115,116,117]. Tindall et al., 2019 [9] found no effect of nut consumption on HbA1c (WMD 0.02%; 95% CI −0.01%, 0.04%; I^2^ = 51.0%). Tindall et al., 2019 [9] observed significant reductions in homeostasis model assessment of insulin resistance (HOMA-IR) (WMD −0.23; 95% CI −0.40, −0.06; I^2^ = 51.7%) and fasting insulin (WMD −0.40 μIU/mL; 95% CI −0.73, −0.07 μIU/mL; I^2^ = 49.4%) after nut consumption from meta-analyses of 19 RCTs [61,66,68,74,86,91,92,96,97,99,100,103,104,105,107,109,110,114,117] and 28 RCTs [58,61,63,66,67,68,74,86,91,92,93,95,96,97,99,100,101,102,103,104,105,107,110,112,113,114,116,117], respectively. 

Lee-Bravatti et al., 2019 [7] conducted a meta-analysis of 9 RCTs [58,66,78,79,80,82,83,86,102] and showed no effect of almond consumption on fasting blood glucose compared with control, and only subjects with CVD risk at baseline showed significant reduction (net change −6.08 mg/dL; 95% CI −10.77, –1.40 mg/dL; I^2^ = 0%, *p*_heterogeneity_ = 0.94) at >42.5 g almond consumption (summary net change −4.11 mg/dL; 95% CI −7.43, −0.80 mg/dL; I^2^ = 34%, *p*_heterogeneity_ = 0.19) compared with controls.

Schwingshackl et al., 2018 [10] conducted a network meta-analysis of 66 RCTs (86 publications with a total of 3595 subjects (280 subjects with T2DM)). They reported nuts are the best food group to lower fasting blood glucose (−0.43, −0.35 mmol/L), compared with refined grains and whole grains.

A meta-analysis by Mazidi et al., 2016 [87] that included 20 RCTs [59,68,74,81,96,98,100,109,110,112,117,118,119,120,121,122,123,124,125,126], and a meta-analysis by Mejia et al., 2014 [89] that included 26 RCTs (healthy subjects [83,117] and subjects with dyslipidemia [65,126], metabolic syndrome [58,66,68,95,99,101,113,116,127,128,129], and T2DM [86,93,94,102,103,112,114,130,131,132]) showed a 0.08 mml/L reduction in fasting glucose. A meta-analysis of 12 RCTs (*n* = 450) by Viguiliouk et al., 2014 [88] showed a 0.15 mmol/L reduction of fasting glucose only in subjects with T2DM.

Overall, nut consumption appears to decrease fasting glucose levels by 0.08 to 0.15 mmol/L compared with control diets.

### 3.10. Blood Lipids

In a previous review [5], nut consumption had been reported to reduce total cholesterol (TC; 0.021 to 0.28 mmol/L) and LDL-C (0.017 to 0.26 mmol/L) compared with control diets from 8 meta-analyses [43,44,87,133,134,135,136,137]. 

In this present review, we included a meta-analysis conducted by Schwingshackl et al., 2018 [10] and Lee-Bravatti et al., 2019 [7]. 

In a network meta-analysis of 66 RCTs (86 publications with a total of 3595 subjects (280 subjects with T2DM)), Schwingshackl et al., 2018 [10] reported nuts are the best food group in lowering LDL-C (−0.34 to −0.24 mg/dL (0.0088 to 0.0062 mmol/L)) and TC (−0.39–−0.30 mmol/L) compared with legumes and whole grains.

Lee-Bravatti et al., 2019 [7] found significant reductions in TC (summary net change = −10.69 mg/dL (0.276 mmol/L); 95% CI −16.75, −4.63 mg/dL; I^2^ = 67%; *p* < 0.01) and LDL-C (summary net change = −5.38 mg/dL (0.139 mmol/L); 95% CI −9.91, −1.75 mg/dL; I^2^ = 61%; *p* < 0.001) with almond consumption in a meta-analysis of 13 RCTs from 14 publications [58,66,78,79,80,81,82,83,85,86,102,138,139,140] including 491 subjects. They [7] observed no difference in triglyceride (TG) levels (summary net change = −11.63 mg/dL (0.1313 mmol/L); 95% CI −23.47, −0.21 mg/dL; I^2^ = 71%; *p* < 0.01) following almond consumption compared with controls in a meta-analysis of 12 RCTs from 12 publications [58,66,78,79,80,81,82,83,86,102,138] including 461 subjects. 

Overall, nut consumption lowers total cholesterol (TC; 0.021 to 0.30 mmol/L) and low-density lipoprotein cholesterol (LDL-C; 0.017 to 0.26 mmol/L) compared with control diets in 10 meta-analyses [7,10,43,44,87,133,134,135,136,137].

### 3.11. Blood Pressure

A previous review [5] showed no effect of nut consumption on blood pressure from meta-analyses of RCTs. In this review, we included a recent meta-analysis conducted by Lee-Bravatti et al., 2019 [7]. They [7] showed no difference (summary net change: −1.51 mm Hg; 95% CI −3.96, −0.94 mm Hg; I^2^ = 49%, *p*_heterogeneity_ = 0.08) between almond consumption and control for diastolic blood pressure (DBP) in the main meta-analysis of RCTs [58,66,78,79,80,81,86], but they found a significant reduction for >42.5 g almond consumption (summary net change: −3.15 mmHg; 95% CI −5.77, −0.54 mm Hg; I^2^ = 35%, *p*_heterogeneity_ = 0.21) and >6 weeks (summary net change: −4.24 mm Hg; CI 95% −6.68, −1.81 mm Hg; I^2^ = 0%, *p*_heterogeneity_ = 0.51). Lee-Bravatti et al., 2019 [7] showed no difference between almond consumption and control for systolic blood pressure (SBP) (summary net change: 1.27 mm Hg; 95% CI −2.63, 5.18 mm Hg; I^2^ = 51%, *p*_heterogeneity_ = 0.07) in summary estimates with almond RCTs [58,66,78,80,81,86,114] compared with control in either the main analysis or subgroup analyses.

### 3.12. Metabolic Syndrome

Zhang et al., 2019 [11] conducted a meta-analysis of 11 observational studies (6 cross-sectional [140,141,142,143,144,145] and 5 prospective studies [146,147,148,149,150]). They found that nut consumption was inversely associated with the risk of metabolic syndrome with RR of 0.84 (95% CI 0.76, 0.92; *p* < 0.001; I^2^ = 79.5%; *p*_heterogeneity_ < 0.001). A subgroup analysis showed this inverse association was only present in tree nuts (RR = 0.97; 95% CI 0.94, 1.00; *p* = 0.04) but not in peanuts (RR = 1.01; 95% CI 0.96, 1.06; *p* = 0.68).

## 4. Discussion

This present review is an updated review of meta-analyses that adds seven more recent meta-analyses aiming to clarify the effect of nut consumption on cardiometabolic disease. We found similar outcomes to our previous review when we combined new and previous meta-analyses investigating CVD mortality, CHD mortality, stroke mortality, CVD incidence, CHD (or CAD) incidence, and stroke incidence. In meta-analyses of interventions, nut consumption significantly reduced fasting glucose levels, TC, and LDL-C compared with controls. However, body weight and blood pressure did not differ after nut consumption compared with controls. 

Schwingshackl et al., 2018 [10] showed the beneficial effect of nut consumption in a network meta-analysis of 66 RCTs (86 publications with a total of 3595 subjects (280 subjects with T2DM)). Nuts ranked highest for LDL-C, TG, TC, HDL-C, fasting blood glucose, HbA1c, SBP, DBP, and C-reactive protein (CRP), in comparison with other food groups, including legumes, whole grains, fish, fruits and vegetables, refined grains, red meat, eggs, dairy, and sugar-sweetened beverages [10]. Clearly, nut consumption appears to decrease cardiometabolic risks.

The results from this present review suggest there needs to be further large clinical trials testing nuts as therapeutic agents for primary and secondary prevention of cardiovascular disease.

## 5. Conclusions

This updated review of meta-analyses found that nut consumption has beneficial effects on cardiometabolic disease with reduced CVD mortality, CHD mortality, stroke mortality, CVD incidence, CHD incidence and stroke incidence comparing high with low categories of nut consumption. It may be attributable to decreases in fasting glucose, total cholesterol and LDL-C.

## Figures and Tables

**Figure 1 ijerph-16-04957-f001:**
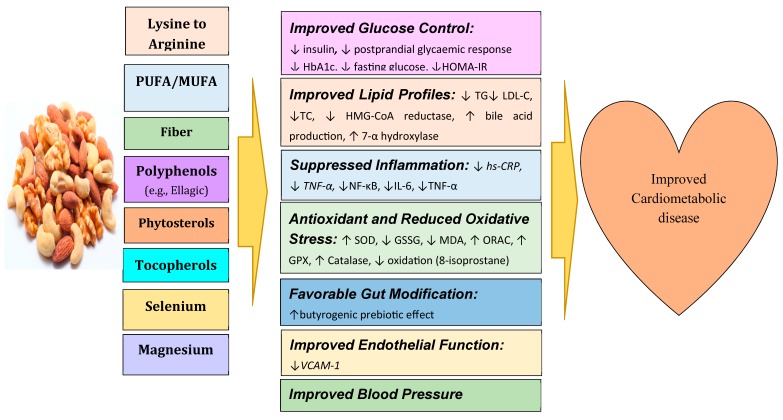
The summary of possible mechanisms linking nut consumption to improved cardiometabolic disease. CCK, cholecystokinin; GLP-1, glucagon-like peptide-1; GPx, Glutathione peroxidase; GSSG, oxidized glutathione; HbA1c, glycosylated hemoglobin; HOMA-IR, homeostasis model assessment of insulin resistance; hs-CRP, high sensitivity C-reactive protein; LDL-C, low density lipoprotein cholesterol; IL-6, interleukin-6; MDA, malondialdehyde; MUFA, monounsaturated fatty acid; ORAC, oxygen radical absorbance capacity; PUFA, polyunsaturated fatty acid; SOD, superoxide dismutase; TC, total cholesterol; TG, triglyceride; TNF α, tumor necrosis factor alpha; VCAM-1, vascular cell adhesion molecule-1; ↑, increase; ↓, decrease.

**Figure 2 ijerph-16-04957-f002:**
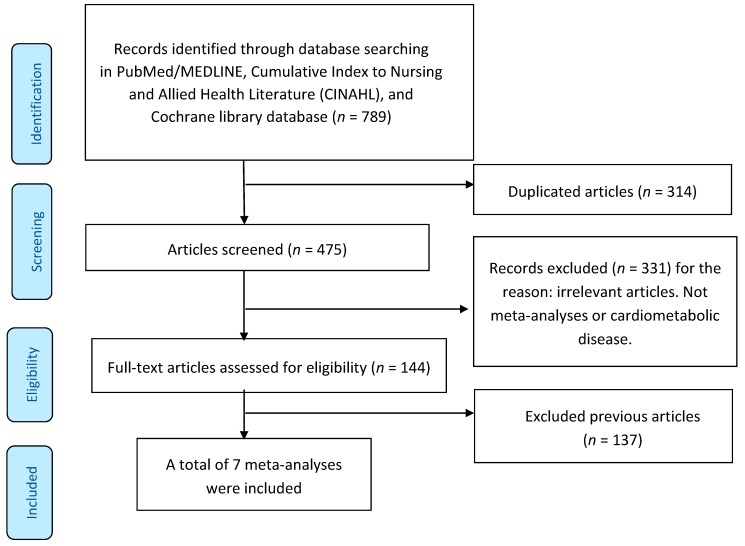
Flow diagram of the literature review.

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
