# Peer review of "Does Nut Consumption Reduce Mortality and/or Risk of Cardiometabolic Disease? An Updated Review Based on Meta-Analyses"

_ijerph, 2019, doi:10.3390/ijerph16244957_

Round 1

Reviewer 1 Report

The authors sought to describe whether nut consumption decreases mortality or/and risk of

cardiometabolic diseases based on updated meta-analyses of epidemiological and intervention studies.

A flow chart depicting the research strategy and selection of studies would be a nice feature.

The instruction could go into a bit more detail on the mechanisms that might be influenced by higher nut consumption in a favorable manner. Also nuts in general could be addresses from a biochemical and its metabolics could be discussed more in-dept.

A schematic figure showing metabolic effects and effects on clinical endpoints could make the paper more appealing.

Author Response

The authors sought to describe whether nut consumption decreases mortality or/and risk of cardiometabolic diseases based on updated meta-analyses of epidemiological and intervention studies.

A flow chart depicting the research strategy and selection of studies would be a nice feature.

Thank you. A flow chart has been created in Line 67.

The instruction could go into a bit more detail on the mechanisms that might be influenced by higher nut consumption in a favorable manner.

Thank you. It has been addressed in lines 32-36 in introduction.

“Nuts comprise 43–67% of fat and 8–22% of protein by weight. Nuts have unique nutritional profiles. Nuts are abundant in unsaturated fatty acids (UFA) (monounsaturated fatty acids (MUFAs) and polyunsaturated fatty acids (PUFAs)) with only 4–5% of saturated fatty acids (SFAs). Nuts are high in vitamins, minerals, fiber and bioactive compounds including carotenoids, phytosterols and polyphenols [1,2].”

Also nuts in general could be addresses from a biochemical and its metabolics could be discussed more in-dept.

Thank you. It has been addressed in lines 37-40 in introduction.

“As we suggested the potential mechanisms of action of nuts [3], MUFA and PUFA are candidates to favourable glucose control and reduced appetite. Arginine and magnesium can contribute to improved inflammation, oxidative stress, endothelial function and blood pressure. Polyphenols can lower risk of type 2 diabetes mellitus (T2DM).”

A schematic figure showing metabolic effects and effects on clinical endpoints could make the paper more appealing.

Thank you. A schematic figure has been included in lines 40-42 in introduction.

“The suggested schematic figure showing metabolic effects and effects on clinical endpoints based on our previous publication [3] are briefly described in Figure 1. “

Reviewer 2 Report

Kim et al. from the Department of Food and Nutrition/Institute of Agriculture and Life Science, Gyeongsang National University, Jinju, Republic of Korea aimed to determine if nut consumption decreases mortality or/and risk of cardiometabolic diseases based on updated meta-analyses of epidemiological and intervention studies. For these purposes they collected 7 new meta-analyses and performed this updated review. Nut consumption significantly decreased cardiovascular disease (CVD) mortality (-19 to -25%; n = 4), coronary heart disease (CHD) mortality (-24 to -30%; n = 3), stroke mortality (-17 to -18%; n = 3) along with CVD (-15 to -19 %; n = 4), CHD (-17 to -34%; n = 8) and stroke (-10 to -11%; n = 6) incidences comparing high with low categories of nut consumption while decreasing blood level of standard risk factors: total cholesterol, LDL and fasting glucose.

This is an informative and interesting and updated (up to November 2019) meta-analysis. However, I suggest to smooth down the conclusion “Nut consumption appears to exert a protective effect on cardiometabolic disease, possibly through improved concentrations of fasting glucose, total cholesterol and LDL-C.” as a cause and effect relationship may not be derived directly from such an analysis. Rather, it should be pointed out that the results encourage to run a really powered new randomized investigation to assess this purposedly and appropriately as so many clues indicate that there might be a potent effect in adding nuts to diet.

Author Response

Kim et al. from the Department of Food and Nutrition/Institute of Agriculture and Life Science, Gyeongsang National University, Jinju, Republic of Korea aimed to determine if nut consumption decreases mortality or/and risk of cardiometabolic diseases based on updated meta-analyses of epidemiological and intervention studies. For these purposes they collected 7 new meta-analyses and performed this updated review. Nut consumption significantly decreased cardiovascular disease (CVD) mortality (-19 to -25%; n = 4), coronary heart disease (CHD) mortality (-24 to -30%; n = 3), stroke mortality (-17 to -18%; n = 3) along with CVD (-15 to -19 %; n = 4), CHD (-17 to -34%; n = 8) and stroke (-10 to -11%; n = 6) incidences comparing high with low categories of nut consumption while decreasing blood level of standard risk factors: total cholesterol, LDL and fasting glucose.

This is an informative and interesting and updated (up to November 2019) meta-analysis. However, I suggest to smooth down the conclusion “Nut consumption appears to exert a protective effect on cardiometabolic disease, possibly through improved concentrations of fasting glucose, total cholesterol and LDL-C.” as a cause and effect relationship may not be derived directly from such an analysis.

Rather, it should be pointed out that the results encourage to run a really powered new randomized investigation to assess this purposedly and appropriately as so many clues indicate that there might be a potent effect in adding nuts to diet.

Thank you. This is addressed in lines 260-261 in discussion.

“The results from this present review suggest there needs to be further large clinical trials testing nuts as therapeutic agents for primary and secondary prevention of cardiovascular disease.”

Round 2

Reviewer 1 Report

The manuscript has improved and all questions were answered.